# Global conservation status of the jawed vertebrate Tree of Life

Rikki Gumbs [1,2,3] ✉, Oenone Scott [3,4], Ryan Bates [1,3], Monika Böhm[5], Félix Forest[6], Claudia L. Gray[1], Michael Hoffmann [1], Daniel Kane[1], Christopher Low[7], William D. Pearse[3], Sebastian Pipins[2,6,8], Benjamin Tapley [1], Samuel T. Turvey [9], Walter Jetz [10,11], Nisha R. Owen[8] & James Rosindell [3]

Human-driven extinction threatens entire lineages across the Tree of Life. Here we assess the conservation status of jawed vertebrate evolutionary history, using three policy-relevant approaches. First, we calculate an index of threat to overall evolutionary history, showing that we expect to lose 86–150 billion years (11–19%) of jawed vertebrate evolutionary history over the next 50–500 years. Second, we rank jawed vertebrate species by their EDGE scores to identify the highest priorities for species-focused conservation of evolutionary history, finding that chondrichthyans, ray-finned fish and testudines rank highest of all jawed vertebrates. Third, we assess the conservation status of jawed vertebrate families. We found that species within monotypic families are more likely to be threatened and more likely to be in decline than other species. We provide a baseline for the status of families at risk of extinction to catalyse conservation action. This work continues a trend of highlighting neglected groups—such as testudines, crocodylians, amphibians and chondrichthyans—as conservation priorities from a phylogenetic perspective.

Human activity threatens to prune entire branches of the vertebrate Tree of Life[1], leading to the loss of billions of years of unique evolutionary history[2]. For mammals alone, it will take millions of years to recover the loss of evolutionary history predicted to occur over the next 50 years[3]. It is critical to avert the greatest losses across the Tree of Life to limit the impact on the capacity of biodiversity to adapt to change and provide benefits to people both now and in the future[4].

The conservation of evolutionary history, typically measured using phylogenetic diversity (PD; the amount of evolutionary history represented by the phylogenetic branches connecting a set of species across the Tree of Life)[5], is linked to the maintenance of evolutionary features, increased ecosystem productivity[6,7], and human

well-being[4,8–10]. Evidence suggests that Amazonian forests that contain greater evolutionary history have higher wood productivity[7], and selecting sets of species to maximise evolutionary history can effectively capture species with known uses by people across the world's plants[4] and birds[11]. As such, PD provides a versatile tool with which to differentiate and prioritise amongst species[12–14] and geographic areas[1,2,15,16] for conservation action. Recognising this, the Intergovernmental Science-Policy Platform on Biodiversity and Ecosystem Services (IPBES) incorporated trends in PD as an indicator to monitor biodiversity's capacity to provide benefits into the future[17]. In addition, the United Nations Convention on Biological Diversity's (CBD) Kunming-Montreal Global Biodiversity Framework (GBF) includes an

[1]Zoological Society of London, London NW1 4RY, UK. [2]Science and Solutions for a Changing Planet DTP, Grantham Institute, Imperial College London, London SW7 2AZ, UK. [3]Department of Life Sciences, Silwood Park Campus, Imperial College London, Ascot, Berkshire SL5 7PY, UK. [4]School of Life Sciences, University of Essex, Colchester CO4 3SQ, UK. [5]Global Center for Species Survival, Indianapolis Zoological Society, Indianapolis, IN 46222, USA. [6]Royal Botanic Gardens, Kew, Richmond, Surrey TW9 3AE, UK. [7]Department of Genetics, Evolution and Environment, Centre for Biodiversity and Environment Research, University College London, London WC1E 6BT, UK. [8]On the Edge, London SW3 2JJ, UK. [9]Institute of Zoology, Zoological Society of London, London NW1 4RY, UK. [10]Department of Ecology and Evolutionary Biology, Yale University, New Haven, CT 06511, USA. [11]Center for Biodiversity and Global Change, Yale University, New Haven, CT 06511, USA. ✉e-mail: Rikki.gumbs@zsl.org

indicator to track the expected loss of PD (hereafter GBF PD indicator) −alongside an indicator to track the changing status of Evolutionarily Distinct and Globally Endangered species (the EDGE Index)[18].

The uptake of evolutionary history as a biodiversity indicator followed IUCN's 2012 resolution (WCC-2012-Res-019)[19] that we must halt the loss of evolutionarily distinct species and lineages−families and orders at risk of extinction−through increased understanding of their status to inform conservation action and policy. However, in the decade following the adoption of the resolution, progress in the global assessment of the conservation status of evolutionary history has largely been limited to terrestrial vertebrates[20–25], sharks, rays and chimaeras (Chondrichthyes)[26], gymnosperms[27] and select invertebrate groups[28,29]. These clades have been prioritised for conservation under the EDGE approach, which ranks species based on their evolutionary distinctiveness and extinction risk[12,14], and regions of the Earth with the greatest concentrations of threatened evolutionary history have been identified[1,2,27,30].

Despite advances in our understanding of threatened evolutionary history for terrestrial vertebrates, we still lack sufficient knowledge for the world's ray-finned fish (Actinopterygii), which together comprise almost half of the world's vertebrates[31] (>32,000 spp.). Recent evidence suggests that ray-finned fish and their habitats are at risk[32] at a global scale, particularly in freshwater ecosystems[33,34]. In addition, for the few small fish clades for which threatened evolutionary history has been assessed, extinction is expected to lead to disproportionately large losses across the fish Tree of Life[35,36].

Here, we generate a global assessment of threatened evolutionary history for the world's jawed vertebrates (Gnathostomata: 70,426 spp.), including an estimate of the conservation status of evolutionary history for ray-finned fish. Our results thus provide data to inform the baseline for the GBF PD indicator[37] across all major jawed vertebrate clades. We also provide an assessment of the 2012 IUCN Resolution to halt the loss of evolutionarily distinct lineages[19] for jawed vertebrates by exploring the current extinction risk and population trends of evolutionarily distinct families. Finally, we applied the EDGE2 framework[14] to identify priority jawed vertebrate EDGE species, including the calculation of EDGE2 scores (hereafter EDGE scores) for ray-finned fish.

## Results

### Global status of jawed vertebrate evolutionary history

Conversions of IUCN Red List categories to quantitative extinction risks are available for various time horizons from 50 to 500 years into the future[3,14,38,39]. We used three quantifications of extinction risk−the 50-year and 500-year time horizons from Mooers et al.[38] and the EDGE2 quantification from Gumbs et al.[14] (see "Methods")−to calculate the total and the proportion of jawed vertebrate evolutionary history (i.e., PD) expected to be lost under different severities of extinction risk.

Together, jawed vertebrates represent 782 billion years (giga-years; GY) of evolutionary history, 42% of which is contributed by ray-finned fish (333 GY), followed by amphibians (143 GY; 18%), which contribute a greater amount than the more speciose birds (Supplementary Fig. 1 and Supplementary Data 1). We currently stand to lose 102 GY of jawed vertebrate evolutionary history (13% of total, based on the EDGE2 quantification of extinction risk[14]; Fig. 1), with estimates ranging from 86 GY (11% of total; 50-year time horizon[38]) to 150 GY (19% of total; 500-year time horizon[38]; all estimates of threatened evolutionary history in Supplementary Data 1). As with total evolutionary history, ray-finned fish, the most speciose vertebrate clade, contribute the largest amount (40% of the total) of threatened evolutionary history at 39 GY (35 GY for 50-year time horizon; 61 GY for 500-year time horizon; Fig. 1a and Supplementary Fig. 1). They are followed by amphibians, which contribute 25 GY of threatened evolutionary history or 26% of the total for all jawed vertebrates (21−37 GY for 50-year and 500-year time horizons, respectively; Fig. 1a), despite contributing just 18% of total evolutionary history.

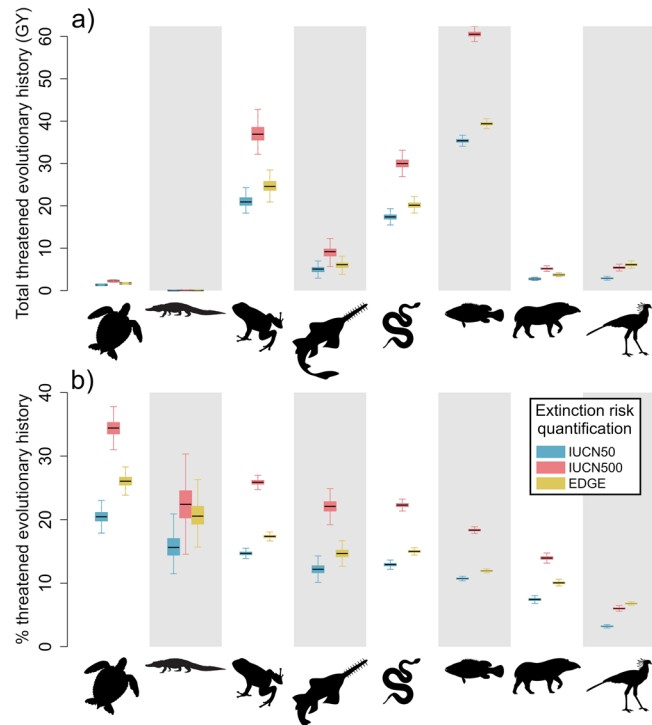

**Fig. 1 | Status of jawed vertebrate evolutionary history.** Threatened evolutionary history of jawed vertebrate clades in (**a**) absolute magnitude (billion years, GY) and (**b**) as a percentage of total evolutionary history of the clade, for three extinction risk quantifications: blue = IUCN50, a 50-year extrapolation of extinction risk from IUCN Red List Criterion E; red = IUCN500, a 500-year extrapolation of extinction risk from IUCN Red List Criterion E; yellow = EDGE, the extinction risk weighting used to generate priority EDGE Lists under the EDGE2 protocol and underpinning the proposed Phylogenetic Diversity indicator for the Kunming-Montreal Global Biodiversity Framework (see "Methods" for extinction risk values used). Boxplot centre line shows the median; box limits, upper and lower quartiles; whiskers show 1.5x interquartile range, calculated across 1000 values for each clade. Clades from left to right: testudines (351 spp.); crocodilians (25 spp.); amphibians (8024 spp.); chondrichthyans (1290 spp.); lepidosaurs (10,735 spp.); ray-finned fish (32,760 spp.); mammals (6253 spp.); and birds (10,988 spp.). Source data are provided as a Source Data file.

In terms of the proportion of evolutionary history at risk across clades, testudines (turtles and tortoises) are at risk of losing 26% of their total evolutionary history under the EDGE2 extinction risk quantification (20%–34% for 50-year and 500-year time horizons), crocodilians 21% (50-year: 16%; 500-year: 22%), and amphibians 17% (50-year: 15%; 500-year: 26%; Fig. 1b). Despite having the greatest total threatened evolutionary history, ray-finned fish are at risk of losing only 12% (50-year: 11%; 500-year: 18%), lower than lepidosaurs (EDGE2: 15%; 50-year: 13%: 500-year: 22%) and chondrichthyans (EDGE2: 15%; 50-year: 12%; 500-year: 22%). Mammals are at risk of losing 10% (50-year: 7%; 500-year: 14%) and birds 7% (50-year: 3%; 500-year: 6%).

### Priority vertebrates for conservation

We calculated the unique evolutionary history (i.e., terminal branch length; TBL) and applied the EDGE2 protocol to generate EDGE scores for 70,426 species of jawed vertebrates (>99% of species) to inform conservation prioritisation[14]. Crocodylian and chondrichthyan species have the highest unique evolutionary history (ANOVA with Tukey Honest Significance Difference: $p > 0.05$ between them, $p < 0.05$ against every other clade; Supplementary Fig. 2).

The amount of evolutionary history embodied by each species in a clade (measured as the total evolutionary history of the clade divided by the number of species) generally decreases as the species richness

a)

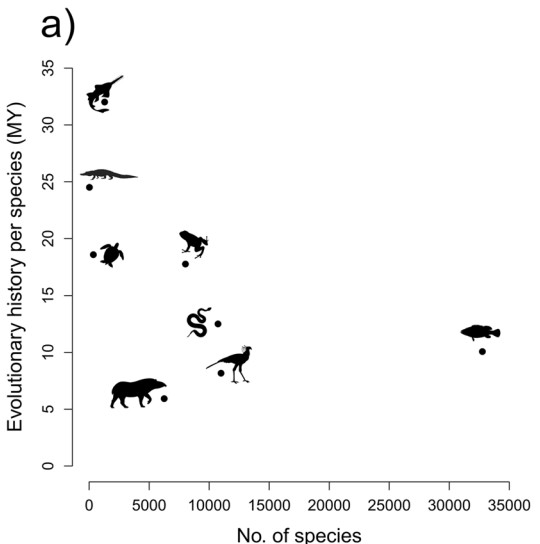
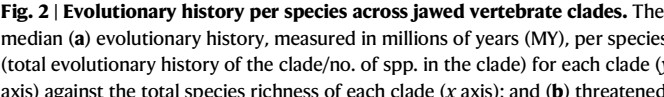

b)

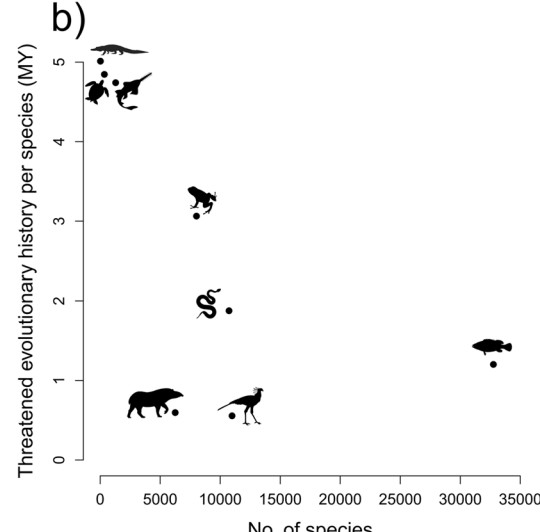

**Fig. 2 | Evolutionary history per species across jawed vertebrate clades.** The median (**a**) evolutionary history, measured in millions of years (MY), per species (total evolutionary history of the clade/no. of spp. in the clade) for each clade (*y* axis) against the total species richness of each clade (*x* axis); and (**b**) threatened evolutionary history per species (total threatened evolutionary history of the clade/no. of spp. in the clade) for each clade (*y* axis) against the total species richness of each clade (*x* axis). Source data are provided as a Source Data file.

of the clade increases (Spearman's rank correlation: $\rho = -0.67$, d.f. = 7, $p = 0.08$; Fig. 2a). Sharks and rays embody the most evolutionary history per species across all jawed vertebrate clades (32 MY per species; ANOVA with Tukey HSD: $p < 0.001$ vs. all other clades; Fig. 2a). They are followed by crocodylians (24.5 MY per species) and testudines (18.6 MY per species). Amphibians embody 17.7 MY of evolutionary history per species despite comprising an order of magnitude more species than testudines, and amphibian evolutionary history per species is more than double that of the similarly speciose mammals (6 MY per species) and more speciose birds (8.2 MY per species; Fig. 2a). Similarly, ray-finned fish embody significantly more evolutionary history per species (10. 1 MY per species) than the much smaller mammal and bird clades ($p < 0.0001$ for all comparisons).

The crocodylians, the least speciose clade in the study, also embody more threatened evolutionary history per species (5 MY; measured as the total threatened evolutionary history of the clade divided by the number of species; Fig. 2b) than any other clade ($p < 0.0001$), followed by the second and third least speciose clades: testudines (4.8 MY per species) and chondrichthyans (4.7 MY per species). Amphibians embody significantly more threatened evolutionary history per species than mammals (3.1 MY vs. 0.6 MY; $p < 0.0001$). Lepidosaurs embody significantly more threatened evolutionary history per species than both mammals and the similarly speciose birds (1.9 MY vs. 0.6 MY; $p < 0.0001$) and the more species-rich ray-finned fish (1.9 MY vs. 1.2 MY; $p < 0.0001$; Fig. 2b).

The two species with the greatest unique evolutionary history diverged from all other extant vertebrates close to the Permian-Triassic extinction event, more than 240 million years ago and are not currently threatened with extinction: the Bowfin (*Amia calva*; 251 MY) and the Tuatara (*Sphenodon punctatus*; 243 MY). The Endangered Salamanderfish (*Lepidogalaxias salamandroides*; 172 MY) has the greatest unique evolutionary history of any threatened species, and 10 of the 25 jawed vertebrates with the greatest unique evolutionary history are threatened with extinction. Three of the five species with the greatest unique evolutionary history are ray-finned fish, and 11 of the top 25 are chondrichthyans (Supplementary Data 1).

We calculated the EDGE scores for each jawed vertebrate species in our analysis using the EDGE2 protocol, which represents the amount of evolutionary history currently at risk that we can expect to safeguard by averting the extinction of a given species[12,14] (see "Methods"). Focusing

conservation attention on species with higher EDGE scores provides priorities to avert the greatest losses of evolutionary history. Testudines and crocodylians have, on average, the highest EDGE scores of all jawed vertebrates, whilst mammals and birds have the lowest (Supplementary Fig. 3). The species with the highest EDGE scores are the Bowmouth Guitarfish (Chondrichthyes: *Rhina ancylostoma*; 88 MY of avertable loss of evolutionary history), followed by the Salamanderfish (Actinopterygii: 84 MY of avertable loss), and the Madagascar Big-headed Turtle (Testudines: *Erymnochelys madagascariensis*; 76 MY of avertable loss; Fig. 3).

Testudines comprise 16% of the top 25 (4 spp.) despite comprising just 0.5% of all jawed vertebrate richness. The top 25 EDGE jawed vertebrates are dominated by amphibians (7 spp., 28% of top 25) and chondrichthyans (6 spp., 24% of top 25), despite these clades representing just 11.4% and 1.8% of total jawed vertebrate richness, respectively, with no mammals or crocodylians present. Four families—Pristidae (chondrichthyans); Sooglossidae and Microhylidae (amphibians); and Eleotridae (ray-finned fish)—feature more than once in the top 25 EDGE species. One of the top 25 EDGE species, the Amani Forest Frog (*Parhoplophryne usambarica*), is Possibly Extinct and another, the Chinese Paddlefish (*Psephurus gladius*), was declared Extinct during the preparation of this manuscript[40] (Fig. 3).

The top 1% of jawed vertebrate species (704 spp.), ranked by EDGE score, capture 15% of total avertable loss of evolutionary history, and the top 9.6% (6760 spp.) capture 50% of total avertable loss. Amphibians comprise 47% (330 spp.) of the top 1% of jawed vertebrate species (Fig. 4a), despite amphibians representing just 11% of jawed vertebrate richness (and 24% of total threatened species richness). Testudines, crocodylians, and chondrichthyans—the three clades with the lowest species richness—are the most overrepresented in the top 1% of jawed vertebrates relative to the number of threatened species in each clade, with 33% (4 spp.) of threatened crocodylians, 33% (53 spp.) of threatened testudines, and 26% (85 spp.) of threatened chondrichthyans in the top 1% (Fig. 4b). There are the same number of chondrichthyans as ray-finned fish in the top 1% (85 spp.) despite the number of threatened chondrichthyans (331) being just 13% of the number of threatened ray-finned fish (2471; Fig. 4b).

There are 52 Possibly Extinct species in the top 1%, and a further three species are potentially lost (terrestrial vertebrates not reliably recorded in past 50 years)[41]. Across the jawed vertebrate Tree of Life,

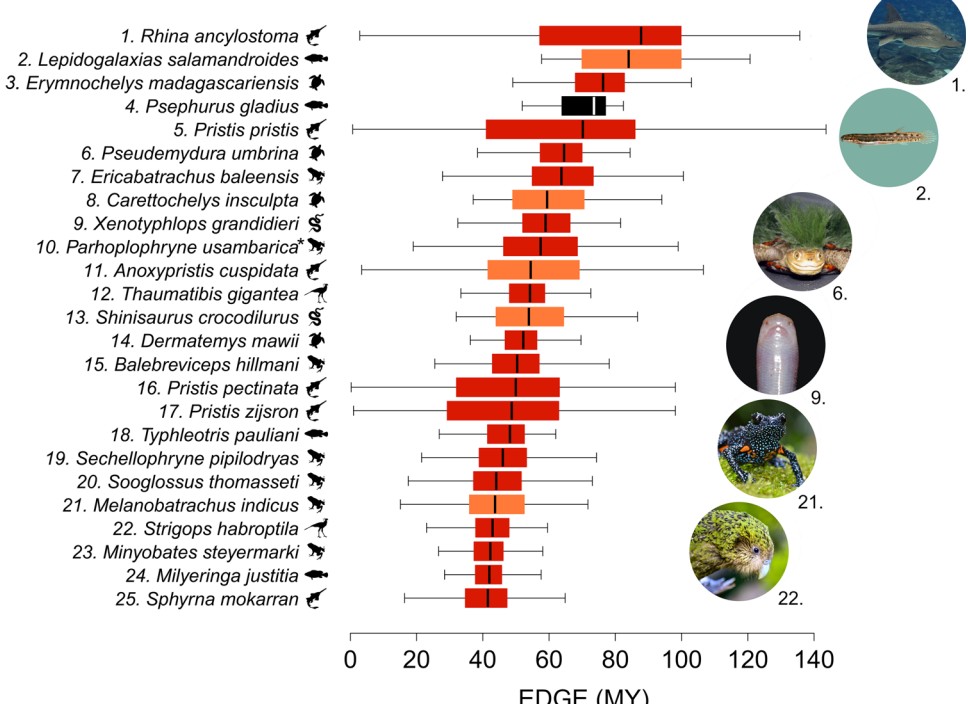

**Fig. 3 | Top 25 Evolutionarily Distinct and Globally Endangered (EDGE) jawed vertebrates.** The 25 jawed vertebrate species with the highest median EDGE scores, calculated from 1000 EDGE scores for each species, generated using the EDGE2 protocol and measuring in millions of years (MY). Colours represent IUCN Red List categories: red = Critically Endangered; orange = Endangered; black = Extinct. *Denotes Critically Endangered species marked with the Possibly Extinct tag on the Red List. Boxplot centre line shows the median; box limits, upper and lower quartiles; whiskers show 1.5x interquartile range. The EDGE score of *Psephurus gladius* (Chinese Paddlefish) was calculated before it was declared Extinct during preparation of the manuscript. Scores and Red List information for all species in this study available in Supplementary Data 1 and Figshare repository (see "Data availability"). Image credits: 1. Brian Gratwicke; 2. Douglass Hoese; 6. Gerald Kuchling; 9. Jörn Köhler; 21. Rajkumar KP; 22. Jake Osborne. Source data are provided as a Source Data file.

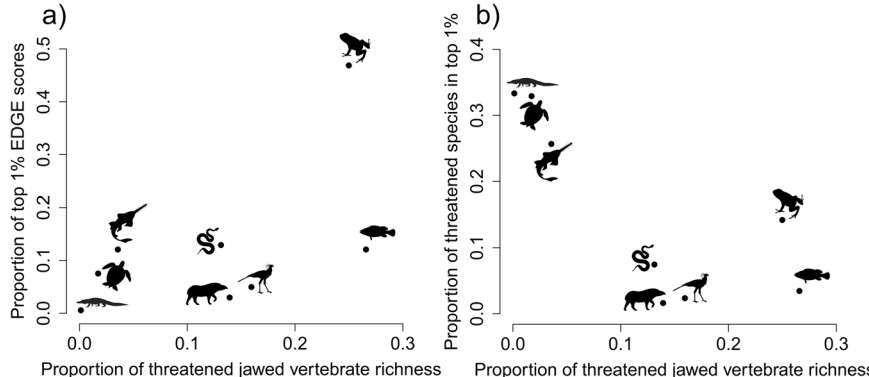

**Fig. 4 | Rankings distribution across clades of the top 1% of jawed vertebrate species according to Evolutionarily Distinct and Globally Endangered (EDGE) scores.** In both panels the points correspond to different monophyletic clades and consider those clades in the context of top 1% EDGE scores of all jawed vertebrates, calculated using the EDGE2 protocol. The *x* axis of both panels corresponds to the proportion of total threatened jawed vertebrate species that are in the relevant clade. The sum of *x* axis values across all data points is therefore one. The *y* axis is different for each panel. In (**a**), the *y* axis gives the proportion of top 1% of jawed vertebrate EDGE-scoring species that are in the relevant clade. The sum of *y* axis values in (**a**) is therefore 1. In (**b**), the *y* axis shows the proportion of threatened species within the relevant clade that are also top 1% jawed vertebrate EDGE-scoring species. Source data are provided as a Source Data file.

we may therefore have already lost 3.4 billion years of evolutionary history that has not yet been confirmed Extinct.

### Evolutionarily distinct lineages

We assessed the extinction risk, current population trends and threatened evolutionary history of jawed vertebrate families to provide important information on the conservation status of evolutionarily distinct lineages as highlighted in the 2012 IUCN Resolution[19]. When considering data sufficient species only (non-Data Deficient assessments on the IUCN Red List), 32 monotypic vertebrate families are fully threatened with extinction excluding any data insufficient species (28% of monotypic families), and 74 families overall (7% of all families), with a mean species richness of polytypic fully threatened families of 6.7 and maximum of 26 (Mammalia: Lepilemuridae). This decreases to 5%

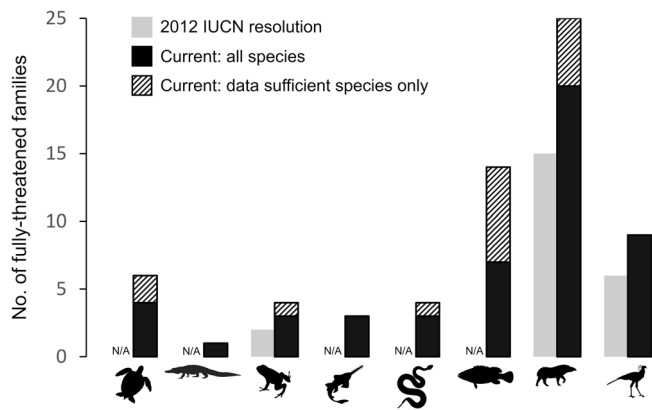

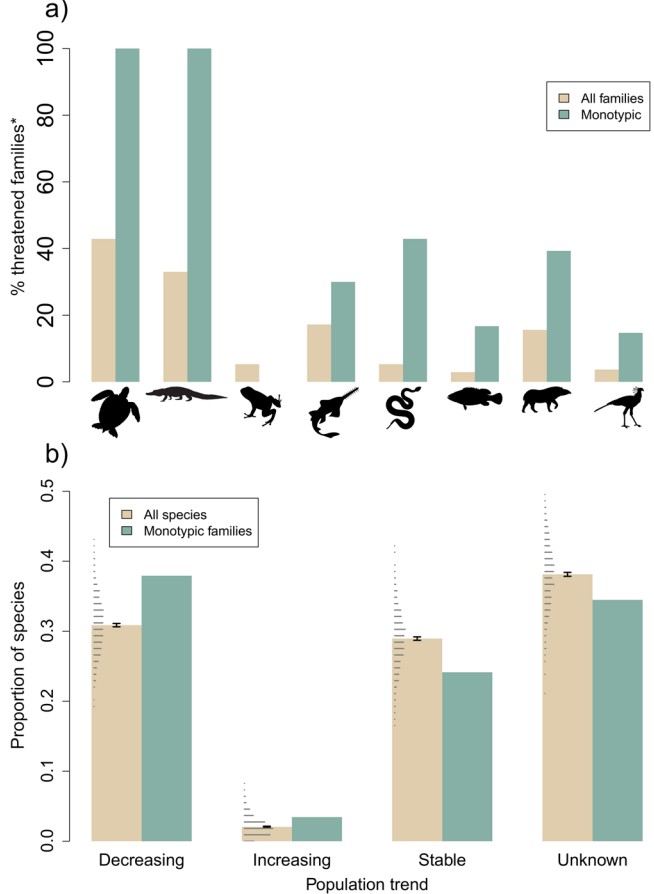

**Fig. 5 | Setting a new baseline for fully-threatened vertebrate families.** The number of fully-threatened families stated in the 2012 International Union for Conservation of Nature (IUCN) Resolution to halt the loss of evolutionarily distinct lineages[19] (grey bars) compared with the number of fully-threatened families reported in this study (black bars). Thatched area of black bars represents the additional families that would be considered fully-threatened if we only include species with data sufficient Red List assessments in the calculation. The IUCN resolution reported values for amphibians, birds and mammals only. Source data are provided as a Source Data file.

of all families and 22% of monotypic ones if we require all species to be data sufficient and threatened (Supplementary Table 1). At the time of adoption of the IUCN Resolution in 2012, every species in 16 mammalian families, six bird families and two amphibian families were threatened with extinction[19]. Our results indicate that this has increased to every species in 20–25 mammal families (depending on inclusion of Data Deficient/unassessed species; Figs. 5 and 6), nine bird families and three to four amphibian families. Taxonomy and extinction risk information alone can be sufficient to reliably identify candidate EDGE priority families (Supplementary Methods) and we also provide the numbers and identity of fully-threatened families for all clades for continued monitoring (Fig. 5, Supplementary Table 1 and Supplementary Data 1).

The proportion of species from monotypic families that are threatened with extinction (22% of all monotypic species, 28% of data sufficient monotypic species) is higher than the observed proportion of threatened species across all jawed vertebrates (13% of all species, 19% of data sufficient species). The 145 monotypic families across jawed vertebrates represent 6 billion years of unique evolutionary history, of which 25% is threatened (1.5 GY from 32 threatened monotypic families). All four monotypic testudine families (Carettochelyidae, Dermatemydidae, Dermochelyidae, Platysternidae) and the single monotypic crocodylian family Gavialidae are threatened with extinction, alongside more than 40% of lepidosaur and mammalian monotypic families (Fig. 6a).

Monotypic families have greater proportions of both decreasing and increasing current population trends than the background expectation for jawed vertebrates, and lower proportions of stable or unknown population trends (Fig. 6b). Four monotypic vertebrate families have increasing populations trends, two of which are non-threatened: Pseudocarchariidae (Crocodile Shark; *Pseudocarcharias kamoharai*; Least Concern) and Pandionidae (Osprey; *Pandion haliaetus*; Least Concern); and two of which are threatened with extinction: Gavialidae (Gharial; *Gavialis gangeticus*; Critically Endangered) and Bolyeriidae (Round Island Keel-scaled Boa; *Casarea dussumieri*; Vulnerable).

To complement the species-level EDGE prioritisation, we calculated the mean EDGE score for each vertebrate family to delineate EDGE Lineages, i.e., the most evolutionarily distinct families where all assessed species are threatened with extinction. These families

**Fig. 6 | Conservation status of evolutionarily distinct lineages. a** The percentage of fully-threatened polytypic families and threatened monotypic families (*considering data sufficient species only) across jawed vertebrate clades. **b** The proportion of monotypic families listed as experiencing different current population trends, compared with a background expectation derived from trends for all jawed vertebrates. Background expectation for (**b**) generated by sampling from all jawed vertebrate species 1000 times and plotted as stacked dots over the bars. Green bars represent monotypic families, whereas beige represent values derived from all families. Bars represent the observed values for (**a**), and observed values for green bars but median values for beige bars in (**b**). Error bars represent 95% confidence intervals (see "Methods"). Numbers of families from (**a**) in Supplementary Table 1. Source data are provided as a Source Data file.

represent deep branches of the Tree of Life that are at the most acute risk and may be overlooked by the species-level approach, as they are descended by multiple species, and this approach can be applied to taxonomic groups where conservation research is often targeted at higher taxonomic ranks (e.g., plants[4]). Overall, the rank of EDGE Lineages is strongly positively correlated with the mean EDGE rank of their constituent species ($\rho = 0.98$, d.f. = 1125, $p < 0.0001$), and the mean species EDGE rank of a family increases (i.e., lower priority) as the species richness of the family increases ($\rho = 0.43$, d.f. = 1125, $p < 0.0001$; Supplementary Fig. 4).

As with species-level scores, testudines and crocodylians have the greatest family-level median EDGE scores of all jawed vertebrate clades (ANOVA with Tukey's Honest Significant Differences: $p < 0.05$ compared with all other clades but $p > 0.05$ between testudines and crocodylians), followed by chondrichthyans and amphibians (Fig. 7). Of the 68 families where all data sufficient species are threatened, 63 have above median family-level EDGE scores and more than 50% assessed species. These 63 families are identified as priority EDGE Lineages (ranked by descending EDGE score in Supplementary Data 1). The top five EDGE Lineages are all monotypic and comprise species present in

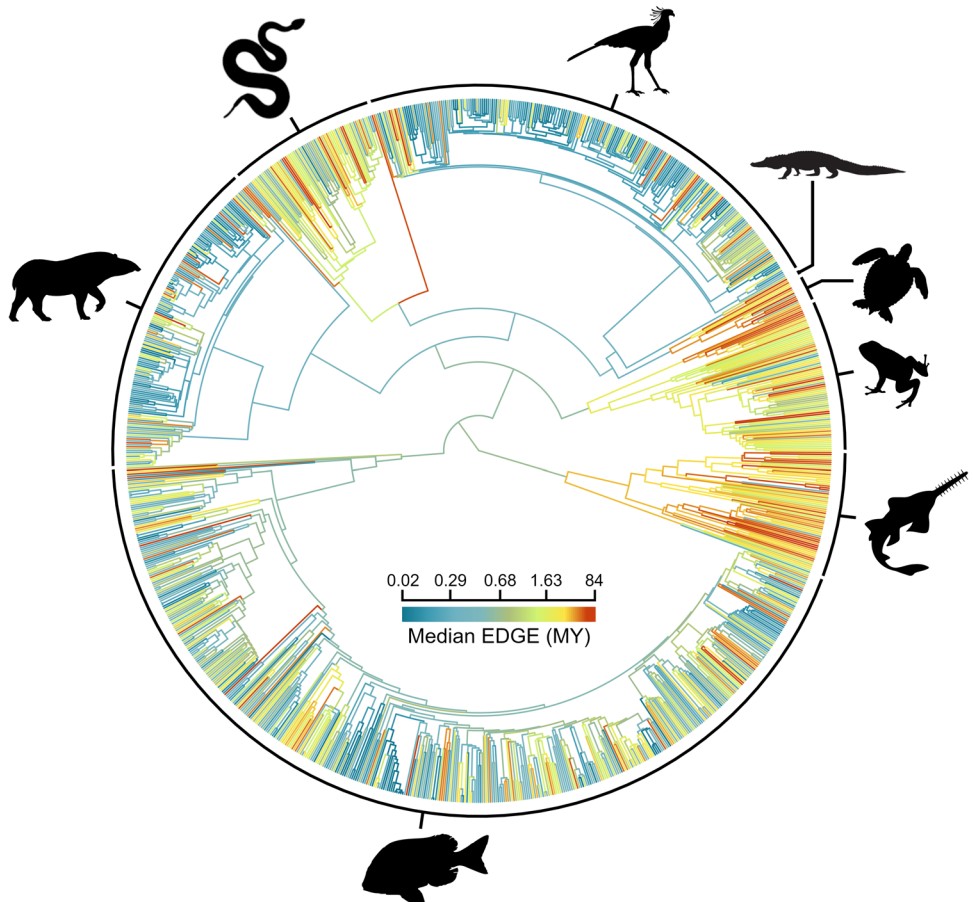

**Fig. 7 | Family-level Evolutionarily Distinct and Globally Endangered (EDGE) scores across the jawed vertebrate Tree of Life.** The median EDGE score for each jawed vertebrate family, with interior branches coloured by the median EDGE score of all species within descendant families, measured in millions of years (MY). EDGE scores calculated using the EDGE2 protocol. Phylogeny constructed for visualisation by stitching together a single constituent phylogeny from the distributions used for each group, with inter-clade node ages taken from the TimeTree of Life[85]. Source data are provided as a Source Data file.

the top 25 EDGE species: the Salamanderfish, the Pig-nosed Turtle (*Carettochelys insculpta*; Carettochelyidae) and Central American River Turtle (*Dermatemys mawii*; Dermatemydidae), and the Madagascar Blindsnake (*Xenotyphlops grandidieri*; Xenotyphlopidae) and the Crocodile Lizard (*Shinisaurus crocodilurus*; Shinisauridae) (full list in Supplementary Data 1).

Despite mammals being absent from the top 25 EDGE jawed vertebrate species and being underrepresented in the top 1% of EDGE scores (Fig. 4), there is a mammal family in the top 25 EDGE Lineages: Daubentoniidae (the Aye-aye, *Daubentonia madagascariensis*). Mammals have the highest number of fully-threatened families of all groups (Supplementary Table 1) and consequently comprise one-third of EDGE Lineages (21 of 63), more than any other clade. The four most speciose EDGE Lineages are primate families: sportive lemurs (Lepilemuridae; 26 spp.); true lemurs, ruffed lemurs and bamboo lemurs (Lemuridae; 21 spp.); gibbons (Hylobatidae; 20 spp.); and sifakas, woolly lemurs and the Indri (Indriidae; 19 spp.). However, mammalian families are significantly younger in age, compared with amphibians, ray-finned fish, chondrichthyans, lepidosaurs and testudines ($p < 0.0001$) and younger, as a proportion of clade age, than families in all other clades excluding ray-finned fish ($p < 0.0001$ vs.; Supplementary Fig. 5).

## Discussion

Here we present a global assessment of the conservation status of the jawed vertebrate Tree of Life. We estimate that more than 100 billion years of evolutionary history is at risk of extinction (Fig. 1a and Supplementary Data 1). Crocodylians and testudines, the smallest clades

with particularly distinctive threatened species, stand to lose the greatest proportions of their evolutionary history (Fig. 1b). Our EDGE prioritisation of jawed vertebrates highlights several evolutionarily unique and threatened ray-finned fish and chondrichthyans as conservation priorities, with the two clades together comprising ten of the top 25 EDGE jawed vertebrates (Fig. 3). In our assessment of evolutionarily distinct lineages, we further find that the proportion of monotypic families threatened with extinction is greater than the background expectation for jawed vertebrates, as is the proportion of monotypic families with decreasing population trends (Fig. 6).

The threatened evolutionary history of tetrapods (56 billion years, or 9%, at risk) is largely congruent with an earlier estimate for the clade that included only 84% of all described tetrapod species (48 billion years of threatened evolutionary history; ~11% of the total evolutionary history)[2]. For birds, the proportion of threatened evolutionary history under the EDGE2 quantification of extinction risk is comparable to that under the pessimistic 500-year scenario. This pattern is driven by the large proportion of Least Concern bird species (77%, the highest of all clades by 25%), which have a higher extinction risk value under the EDGE2 quantification[14] (0.06) than the 500-year quantification (0.0005), and the relatively small proportion of EN (4%) and CR (2%) species, which have lower extinction risk values under the EDGE2 quantification (0.485 and 0.97 vs. 0.996 and 1 under the 500-year quantification[38], respectively). Estimations of threatened evolutionary history using the EDGE2 quantification are therefore more pessimistic for LC species, relative to the 500-year quantification, but more optimistic for EN species.

Our estimate of threatened evolutionary history for jawed vertebrates using current extinction risk data provides a global baseline for this diverse and ecologically significant clade, and informs the Phylogenetic Diversity indicator adopted by the UN's Kunming-Montreal GBF[42]. Our findings indicate that earlier quantifications of threatened evolutionary history for IPBES assessments[43] likely overestimated the magnitude of threatened evolutionary history for birds (12.5% vs. 3%–7% here; Fig. 1b) and mammals (26% vs. 7%–14% here; Fig. 1b), and underestimated the threatened evolutionary history for lepidosaurs (8% vs. 13%–22% here; Fig. 1b). This incongruence is to be expected given that the initial IPBES approach does not consider the potential extinction risk of Data Deficient or unassessed species (37% of jawed vertebrates), does not differentiate between the extinction risk of threatened species, and does not incorporate the interplay of extinction risk on internal branches of the Tree of Life[44]. We thus recommend that future IPBES assessments adopt the approach used here to estimate threatened evolutionary history, for improved robustness and consistency with the GBF PD indicator.

The evolutionary history of testudines and crocodylians is at particularly acute risk (Figs. 1 and 2b). This is a consequence of both high overall extinction risk and the large proportion of total and threatened evolutionary history represented by a small number of species, and is exacerbated by the concentration of threatened species on long terminal branches (Supplementary Fig. 2). Despite having marginally lower levels of overall threatened evolutionary history than crocodylians or testudines, amphibians and sharks and rays embody particularly large amounts of threatened evolutionary history relative to their species richness (Fig. 2b), indicating the loss of species from these clades will likely result in larger losses of evolutionary history on average. Lepidosaurs have a proportion of threatened evolutionary history comparable to chondrichthyans, despite having a smaller proportion of threatened species, due to the large amount of total and threatened evolutionary history embodied by each lepidosaur species on average, which is significantly greater than that of the comparably speciose birds (Fig. 2). This large amount of per-species evolutionary history embodied by lepidosaurs, relative to similarly speciose groups, is driven by their ancient crown age (~240 MYA, compared with ~150 MYA for birds), relatively old family-level phylogenetic divergences (Supplementary Fig. 5), and relatively long terminal branches (Supplementary Fig. 2).

Furthermore, our approach to incorporating Data Deficient and unassessed species considers their extinction risk to be uncertain but overall comparable to that of Vulnerable species. This leads to clades with notable numbers of Data Deficient or unassessed species, such as lepidosaurs and ray-finned fish, to effectively have increased proportions of threatened species in our analyses relative to the proportion currently observed for the clade. Previous estimates give support to our approach of taking a pessimistic view of data insufficient species[45]. Given the importance of including Data Deficient species in estimates of threatened evolutionary history[2,46,47], future calculations may go further and explicitly incorporate predictions of whether poorly-known species are threatened or not to parameterise the extinction risk values they are assigned, and provide more precise estimates of the expected loss of evolutionary history[45,48].

Aquatic species dominate the highest priority EDGE ranks, with seven of the top 10 and 14 of the top 25 jawed vertebrates occurring in freshwater or marine habitats (Fig. 3). This reflects the plight of aquatic vertebrate species[32,33,49] and their evolutionary history[24,26,35,36], and the lack of conservation attention aquatic species relative to their terrestrial counterparts[50–52]. Indeed, despite comprising relatively few species, the long terminal branches of threatened crocodylians, testudines, and chondrichthyans—compared with other vertebrate clades (Supplementary Fig. 2)—mean these three clades have the largest proportions of their threatened species present in the top 1%. Similarly, the over-representation of amphibians in the top 1% of EDGE scores reflects not only the high evolutionary uniqueness of amphibian species[23], but also the much greater number of threatened amphibians (2323 species) relative to other vertebrate clades[53]. This is second only to the number of threatened ray-finned fish (2471; Fig. 4a), despite amphibians totalling just a quarter of ray-finned fish species richness.

Worryingly, 17% of all Critically Endangered jawed vertebrates are marked as Possibly Extinct. The imminent loss of these species and their evolutionary history is very real: during the preparation of this manuscript, the fourth-highest-ranking jawed vertebrate, the Chinese Paddlefish (*P. gladius*), was declared extinct[40]. Its extinction signals the loss of an entire genus and a branch of the Tree of Life that originated shortly after the Cretaceous-Paleogene boundary; the unique evolutionary history embodied by the Chinese Paddlefish (63 MY) was amongst the top 0.02% for all jawed vertebrate species (Supplementary Data 1). This extinction represents not only a significant loss of evolutionary history but also of ecological diversity: the Chinese Paddlefish was amongst the largest freshwater fish[40] and utilised passive electroreception to locate its fish prey[54]. This extinction leaves the deep branch connecting the family Polyodontidae to all other extant life in a precarious position; the single remaining polyodontid species, the American Paddlefish (*Polyodon spathula*), is currently listed as Vulnerable on the Red List and is now solely responsible for more than 120 million years of unique evolutionary history.

The extinction of the Chinese Paddlefish represents the latest stage of the degradation and erosion of evolutionarily distinct endemic biodiversity once supported by China's Yangtze River system, following the recent loss of the Yangtze River Dolphin or Baiji (*Lipotes vexillifer*), a monotypic endemic mammal family (Lipotidae), from the same system[55]. Our results also highlight the increased vulnerability of other specific geographic regions to the loss of disproportionately high levels of unique evolutionary history[1,2]. For example, all representatives of the fully threatened jawed vertebrate family with highest species richness (Lepilemuridae, 26 species) occur in Madagascar, which is also home to one of the species with the overall highest EDGE scores (Madagascar Big-headed Turtle). This spatial concordance of progressive or imminent loss of high-ranking vertebrates from particular ecosystems highlights the need to consider our findings within a spatial framework in the future, to better understand the relationship between geographic range, ecology and extinction risk for particularly distinctive species[56], and specifically to identify top-priority landscapes and regions that require urgent attention to prevent the extinction of large amounts of unique evolutionary history. Such losses may be particularly likely in aquatic systems that have not previously been assessed for conservation priority in terms of unique evolutionary history, due to the lack of comparative data for ray-finned fishes.

We do not appear to be on track to halt the loss of evolutionarily distinct lineages[19]. Our findings instead suggest that the conservation status of evolutionarily distinct jawed vertebrate lineages has not improved since 2012, with a greater number of amphibian, mammal and bird families now fully threatened with extinction (34 cf. 24 in 2012; Fig. 5 and Supplementary Table 1). The conservation status of monotypic families is actually worse than expected: a higher proportion of monotypic families are threatened with extinction, and monotypic families have a significantly greater number of declining population trends than the background expectation for jawed vertebrate species in general (Fig. 6). Monotypic families at risk include some of the most ecologically distinctive birds and mammals, such as the Hihi (*Notiomystis cincta*), Kagu (*Rhynochetos jubatus*), Aye-aye and Red Panda (*Ailurus fulgens*; all species in the top 10% of functional distinctiveness for their clade, according to Pollock et al.[16]).

Our approach to identifying priority lineages at the family level provides an alternative to species-level EDGE scores that highlights particularly deep phylogenetic branches at risk due to the acute

extinction risk of both distinct monotypic lineages and families comprising multiple species Such a prioritisation at a higher taxonomic level is more robust to the impact that taxonomic lumping and splitting and data deficiency can have on their prioritisation[57,58]. The high proportion of mammalian families comprising jawed vertebrate EDGE lineages—despite their relative absence from the top 1% of species EDGE scores (Fig. 3)—suggests that while individual mammal species may not represent incredibly large amounts of threatened evolutionary history, the phylogenetic clumping of extinction risk across entire families is nevertheless threatening deep branches of the mammal phylogenetic tree. It is for this reason that EDGE species—the most evolutionarily distinct and threatened species from a given clade identified to guide priority setting—are identified independently for each vertebrate class[12,21]; to ensure clade-specific priorities for conserving evolutionary history are not overlooked when focusing on the wider Tree of Life. This becomes ever more important as EDGE scores are calculated for across an increasing number of clades across the entire Tree of Life. Whilst we present all jawed vertebrate species combined here to highlight priorities across the entire jawed vertebrate Tree of Life, we also determined whether each species is an EDGE species within its respective clade to guide priority setting (Supplementary Data 1). EDGE Lists for all clades are maintained and updated by the Zoological Society of London's EDGE of Existence (www.edgeofexistence.org) to support conservation decision making.

Whilst we here set strict criteria for EDGE lineages, we recognise that this is reasonable only for a select few clades with exceptional Red List coverage. Future EDGE lineage calculations for clades lacking such comprehensive Red List data could therefore explore lower or multiple thresholds for determining priority lineages (e.g., >50% of species in the lineage are threatened, or above the overall observed proportion of threatened species for the higher taxonomic unit being assessed) and utilise the predictions of machine learning or other non-Red List approaches to inform the overall extinction risk of a family for ranking lineages[45,48].

Despite the worsening status of evolutionarily distinct lineages, the IUCN Red List documents current population trends as increasing for two monotypic threatened families, the Gharial[59] and the Round Island Keel-scaled Boa[60]. The ongoing recovery of both species indicates that conservation can bring species back from the brink of extinction. Both species have been subject to targeted conservation actions following their population falling below 100 or so adult individuals. These conservation actions have averted even greater losses for both species and resulted in improved outlooks, including the recent downlisting of the boa from Endangered to Vulnerable on the IUCN Red List[60]. If we are to halt the further loss of evolutionarily distinct lineages, targeted conservation action is required for a greater number of species. Indeed, the species included in our current analysis still represent just ~1% of life on Earth[61]. As advances are made to estimate the wider Tree of Life[27,29,62–64] and to fill gaps in our understanding of extinction risk for groups such as plants[65–67], our approach for estimating threatened evolutionary history with incomplete data provides an avenue to incorporate clades currently precluded from such analyses due to lack of data. In addition, indicators such as the EDGE Index, designed to track the status of evolutionarily distinct species through time, provide ways to monitor our progress towards conserving the most evolutionarily distinct species[68].

The sixth mass extinction has already begun pruning deep branches of the Tree of Life, and we have shown that we are at risk of losing more than a hundred billion years of evolutionary history without urgent action (Fig. 1). Numerous evolutionarily distinct species, many of which are the sole surviving species of entire lineages, are heading towards extinction despite continued recognition that they require dedicated conservation attention. We must therefore better integrate these species into conservation planning, policy, and action. If we fail,

we risk losing many of the most distinctive and irreplaceable species on Earth, isolated on some of the Tree of Life's longest branches, along with their associated evolutionary features, ecological novelties, and benefits to people.

# Methods

## Data

For our analysis of jawed vertebrate evolutionary history, we included eight major monophyletic clades—amphibians; birds; crocodylians; lizards, snakes and the Tuatara (Lepidosauria); mammals; ray-finned fish (Actinopterygii); sharks, rays and chimaeras (Chondrichthyes); and turtles and tortoises (Testudines)—which have strong phylogenetic and extinction risk data coverage and together account for >99% of jawed vertebrate species. For amphibian taxonomy, we used Frost's Amphibian Species of the World[69], which included 8024 species. We matched this taxonomy to the phylogeny of Jetz and Pyron[23], for which we retained 7002 species. For birds we used BirdLife International's Handbook of the Birds of the World v5.0[70], identifying 10,988 valid and extant species, and matched the taxonomy to the phylogeny of Jetz et al.[20], retaining 9645 species. For mammals we used the mammal taxonomy and phylogenetic trees from Gumbs et al.[14], which adopted the Mammal Diversity Database v1.1[71] of the American Society of Mammalogists, identifying 6253 extant and valid mammals species, and matched the taxonomy to the phylogeny of Upham et al.[72], retaining 5853 species. For chondrichthyans we used Fishbase[73], which included 1290 species, and matched the taxonomy to the phylogeny of Stein et al.[26], retaining 1165 species. For ray-finned fish we also used Fishbase[73], which comprised 32,760 species, and matched this taxonomy to the phylogeny of Chang et al.[74], retaining 31,506 species. For crocodylians, testudines and lepidosaurs, we used the Reptile Database[75], which included 25, 351, and 10,735 species, respectively. We matched the reptilian taxonomy to the phylogeny of Colston et al.[24] for crocodylians and testudines, retaining 25 crocodylian and 325 testudine species, and to the phylogeny of Tonini et al.[22] for lepidosaurs, retaining 9599 species.

For all clades except mammals we randomly sampled 1000 trees from the available distributions to adequately capture uncertainty in phylogenetic relationships and node ages[20,76]. To maximise the inclusion of species, and enable effective comparison between clades, we imputed species missing from the phylogenies, following taxonomy matching, to generate 1000 phylogenetic trees for each clade that comprised all valid species present in each taxonomic treatment (1022 amphibians, 1343 birds, 1691 ray-finned fish, 135 chondrichthyans, 26 testudines and tortoises, and 1136 lepidosaurs). For imputation we followed earlier approaches to insert missing species into their genus along the existing phylogenetic branches[14,27,77], using the *congeneric.impute* function in the R package *pez*[78]. This approach has been used to provide estimates of threatened phylogenetic diversity[77] and was used to generate the mammal trees used here, which have been shown to produce robust species prioritisations[14,27]. For mammals, we used the imputed trees of Gumbs et al.[14], which included the 400 missing species and utilised the same imputation approach applied here. This produced 1000 sets of phylogenies for each of the eight clades, together comprising 70,426 jawed vertebrate species. We used extinction risk data for all species available from the IUCN Red List of Threatened Species version 2021.1[53], for which there were 44,530 data sufficient assessments (Least Concern [LC], Near Threatened [NT], Vulnerable [VU], Endangered [EN], Critically Endangered [CR] and Extinct in the Wild [EW]) and 3273 Data Deficient assessments.

## Global status of jawed vertebrate evolutionary history

To estimate the global status of jawed vertebrate evolutionary history, we calculated the amount of evolutionary history we currently stand to lose, given the extinction risk of species within each clade[44,79]. To produce the total evolutionary history of a clade, we calculated the

total phylogenetic diversity (PD) by summing all branch lengths of the phylogenetic tree:

$$PD(k) = \sum_{i=1}^{b} L_i \tag{1}$$

where $k$ represents the phylogenetic tree, $b$ gives the total number of phylogenetic branches in $k$, $i$ is an index identifying individual branches from $k$, and $L_i$ is the length of branch $i$ where $1 \le i \le b$. To estimate the amount of evolutionary history at risk, we follow Faith[79] by calculating the expected PD loss:

$$\text{expected PD loss}(k) = \sum_{i=1}^{b} L_i \left( \prod_{j=1}^{n_i} q_{ij} \right) \tag{2}$$

where $n_i$ gives the number of descendant species from branch $i$, and $q_{ij}$ gives the probability of extinction for descendant species $j$ from branch $i$ where $1 \le i \le b$ and $1 \le j \le n_i$. This approach calculates the amount of evolutionary history from each phylogenetic branch we expect to lose, given the extinction probability of all descendant species of that branch. For example, if we consider a branch descended by two species, both of which have a probability of extinction of 0.5, there is a 0.5*0.5 (0.25) probability of losing the branch's evolutionary history, and thus the expected loss is 25% of the branch's evolutionary history. When this is calculated and summed for all branches in the phylogenetic tree, we have the total amount of evolutionary history that we stand to lose for the entire clade, and from this we can derive the proportion of total evolutionary history expected to be lost. We calculated the percent of evolutionary history at risk as:

$$\text{Prop. thr. evo. hist.}(k) = \frac{\text{Expected PD loss}(k)}{PD(k)} \times 100 \tag{3}$$

This proportion of the total evolutionary history of a clade we expect to lose underpins the Phylogenetic Diversity indicator adopted by the CBD's GBF[18,37].

We calculated the amount of evolutionary history at risk of being lost for all 1000 trees of each clade under three existing extinction risk weightings onto which we mapped the IUCN Red List categories of species: the IUCN50 and IUCN500 extinction risk weightings[38], which represent conversions of IUCN Red List categories to probability of extinctions of today's threatened species at 50 years and 500 years into the future, respectively; and the EDGE2 weighting, which is used to identify priority EDGE species for conservation[14] and underpins the PD and EDGE indicators included in the GBF[18,37] (Table 1).

We incorporated uncertainty around extinction risk in two ways, following the EDGE2 prioritisation framework[14]: (1) we generated a distribution of extinction risk values associated with each Red List category derived from a fitted curve, for which the median of each set of values associated with a given Red List category aligned with its original IUCN50, IUCN500, or EDGE2 value; and (2) for each iteration, Data Deficient (DD) and Not Evaluated (NE) species had their extinction risk weighting selected from the entire distribution of extinction risk values at random, generating an uncertain distribution of scores with a median equivalent to the Vulnerable Red List category. Thus, during the calculation of threatened evolutionary history on any given phylogenetic tree, DD and NE species have a 60% chance of being assigned an extinction risk weighting associated with one of the three threatened Red List categories (VU, EN, CR) and a 40% chance of being assigned an extinction risk weighting associated with the two non-threatened Red List categories (LC, NT). This proportion is comparable to some predictions that 59% of jawed vertebrates may be at risk of extinction[45].

**Table 1 | Quantifications of extinction risk**

| Red List category | | IUCN50 | IUCN500 | EDGE2 |
|---|---|---|---|---|
| Data deficient and unassessed | Least concern | 0.00005 | 0.0005 | 0.060625 |
| | Near threatened | 0.004 | 0.02 | 0.12125 |
| | Vulnerable | 0.05 | 0.39 | 0.2425 |
| | Endangered | 0.42 | 0.996 | 0.485 |
| | Critically endangered | 0.97 | 1 | 0.97 |

The quantitative conversions used to translate IUCN Red List categories to probabilities of extinction to estimate threatened evolutionary history. IUCN50 and IUCN500 are from Mooers et al.[38] and "EDGE" is the conversion used in the EDGE2 calculation from Gumbs et al.[14]. For all, Data deficient and unassessed species are drawn at random from all possible values.

An alternative approach to incorporating DD and NE species would be to draw their extinction risk weighting species in proportion with the observed distribution of Red List categories for a given clade. However, the extent to which extinction risk is dispersed−or clumped−on the Tree of Life varies between and within clades[80–82]. Using clade-specific parameters to draw the extinction risk values of DD/NE species would lead to significant inter-clade variation in the impact of DD/NE species on the weighting of internal branches for calculations of threatened evolutionary history, further compounded by the different proportion of DD/NE between clades. Further, DD/NE terrestrial vertebrate species are likely to be similarly range-restricted and human-impacted to EN and CR species[2], and more than half of DD species are likely threatened[45]. Thus, using the observed distribution of extinction risk for all jawed vertebrates, whether clade-specific or pooled for the entire group, to parameterise the extinction risk values of DD/NE species would significantly underestimate their extinction risk (e.g. the observed proportion of threatened amphibians is ~40%, whereas 85% of DD species are predicted to be threatened, and the observed proportion of threatened jawed vertebrates is ~19% whereas 59% of DD species are predicted to be threatened)[45].

We used terminal branch lengths as a measure of minimum evolutionary distinctiveness (ED) that is insensitive to the choice of extinction risk weighting on internal branches. We calculated the median terminal branch length for each species across the 1000 phylogenies of their respective clade. To compare terminal branch length between clades, and for all subsequent cross-clade analyses, we used ANOVA with Tukey's Honest Significant Difference test for pairwise comparison. We used Welch's $t$-test to compare the terminal branch lengths of threatened and non-threatened species within each clade.

**Priority vertebrates for conservation**

We calculated the amount of evolutionary history currently at risk of being lost for each species using the EDGE2 protocol, which sums the threatened evolutionary history associated with each phylogenetic branch ancestral to a given species and can be derived from the extinction risk-transformed trees generated to calculate the clade-level expected loss of evolutionary history outlined above[14,37,83]. For each species, we calculated the median EDGE score from the distribution of 1000 extinction risk-transformed phylogenies. To estimate the unique evolutionary history of each species, we calculated its median terminal branch length from the 1000 phylogenetic trees. For each species we also calculated the amount of evolutionary history the species is expected to be responsible for in the future, given the extinction risk of all other species in the tree, previously referred to as Heightened ED[83] or ED2 scores[14]. To ensure these findings are immediately applicable for conservation action, we used the EDGE2 extinction risk weightings to derive EDGE scores for all species, as this approach underpins both the EDGE Lists produced for all clades by the Zoological Society of London's EDGE of Existence programme, and the PD-related indicators

included in the Kunming-Montreal GBF[14,18,37]. We followed Gumbs et al.[14] by calculating EDGE2 scores for species $i$ as:

$$EDGE2_i = ED2_i \times GE2_i \qquad (4)$$

where the EDGE2 of species $i$ is a product of its ED2 and GE2 scores. GE2 scores are the extinction risk weightings outlined in Table 1, and ED2 is calculated for species $i$ as:

$$ED2_i = TBL_i + \sum_{j \in A(i)} \left( L_j \times \prod_{h \in C_j\backslash\{i\}} GE2_h \right) \qquad (5)$$

where $ED2_i$ consists of the $TBL_i$ (terminal branch length of species $i$) plus another component corresponding to the expected contribution of shared internal branches to future terminal branch length $i$. The set $A(i)$ contains all branches ancestral to species $i$ but excluding the terminal branch. As before, $L_j$ gives the length of branch $j$. The set $C_j$ contains all descendant species of internal branch $j$. $C_j\backslash\{i\}$ uses mathematical set minus notation to indicate contains all descendant species of internal branch $j$ excluding species $i$. $GE2_h$ is the extinction risk weighting of species $h$. $ED2_i$ is therefore species $i$'s expected terminal branch length at some future time after all other species in the tree survive or become extinct based on their associated $GE2_h$ values (i.e., product of all $GE2_h$)[14].

To identify priority jawed vertebrates whose conservation would capture large amounts of threatened evolutionary history, we ranked all species by their EDGE score in descending order. We arbitrarily selected the top 1% (704 spp.) to represent our priority set of species for further exploration[84]. To explore the proportional representation of clades in the top 1%, we calculated: (1) the proportion of the top 1% species comprised by each clade; (2) the proportion of each clade's total richness present in the top 1%; and (3) the proportion of threatened richness of each clade present in the top 1%. We also identified all species that are listed as Possibly Extinct on the IUCN Red List for all clades, and also identified the lost taxa identified by Martin et al.[41] for tetrapods to estimate the number of priority EDGE species that are possibly extinct or likely to be lost, by calculating the evolutionary history that is likely to have been lost with the extinction of all lost or possibly extinct species. For this, we used the conservative estimate of summing the terminal branch lengths for all lost or possibly extinct species. We also applied the EDGE2 criteria to identify sets of priority species for conservation in each clade (EDGE species): above median EDGE for the clade in 95% of calculations and in a threatened IUCN Red List category to guide conservation action (full list of species and their EDGE scores in Supplementary Data 1).

## Evolutionarily distinct lineages

Following IUCN's 2012 resolution to halt the loss of evolutionarily distinct lineages[19], there has been, to our knowledge, no subsequent direct assessment of the status of the lineages considered of high conservation importance. Here, as per the IUCN resolution, we focus our assessment on threatened monotypic families and families in which all species are threatened as of utmost conservation importance to avoid large losses of irreplaceable biodiversity. To provide an update on the status of these families, for each clade we estimated two values: (1) the number of families for which all species are threatened and (2) the number of families for which all assessed species are threatened. For amphibians, birds and mammals it was possible to contrast this with the number of threatened families in 2012 as all three groups had been comprehensively assessed. Our estimates for the remaining clades represent the first assessment of evolutionarily distinct lineages, to our knowledge.

For monotypic families we calculated the proportion of species listed as having decreasing, increasing, stable or unknown population

trends on the IUCN Red List and compared this with a null expectation. For the null expectation we selected random sets of species that matched those in monotypic families (both in clade affinity and Red List category) from the pool of all species with available population trend data. In other words, the sets of species comprising the null distribution are equal to the set of monotypic families in terms of proportion of species belonging to each clade and to each Red List category. We repeated this to generate a distribution of 1000 sets of trends representing the background expectation for jawed vertebrates. We compared the numbers of species listed under each trend category for monotypic families and all jawed vertebrates using one-sampled $t$-tests, with the observed number of monotypic species in each category as the value of the mean against which to compare the null distribution.

To calculate a family-level EDGE score, we calculated the mean EDGE score for all species within the family and assigned that to the family. We hereby define EDGE Lineages as the families for which: (1) all data sufficient species are threatened; (2) the family-level EDGE score is above median for all families; and (3) at least 50% of species in the family have been assessed in a data sufficient category on the IUCN Red List. We calculated the mean, rather than the median, EDGE score for each family as the mean EDGE score in this case would represent the average amount of threatened evolutionary history we could expect to conserve with conservation action on a random species in the family. In practice, there was minimal difference between the use of median vs. mean EDGE scores to characterise EDGE at the family level (>98% similarity in families selected as EDGE lineages).

We also calculated the overall age of each family as the median stem age (i.e., the distance from the tip of the tree to the most recent internal node shared by the family and its sister) across all phylogenetic trees. We took this measurement, rather than the crown age of families, for consistent inclusion of monotypic families, which lack an analogous crown age. For each clade and all jawed vertebrates combined, we then calculated the number and proportion of monotypic families, the number and proportion of fully-threatened families, and the number and proportion of families in which all data sufficient species are threatened.

## Using taxonomy to identify candidate EDGE species

Currently, the identification of priority EDGE species is limited to groups for which extensive phylogenetic and IUCN Red List data are available. However, this is currently the case for a small fraction of the entire Tree of Life. We used taxonomic and Red List information with the view of determining criteria for identifying candidate EDGE species for conservation action from clades lacking the phylogenetic data to conduct comprehensive EDGE assessments (i.e. ~ 99% of the Tree of Life[61]). To do this we calculated the proportion of species in each clade that: (1) are in monotypic or fully-threatened families; and (2) are Vulnerable, Endangered, Critically Endangered or Extinct in the Wild on the IUCN Red List (i.e. threatened); and also meet the EDGE species criteria (above median EDGE for the clade in 95% of calculations and in a threatened IUCN Red List category)[14,21]. This would then tell us whether threatened species from either monotypic families or families where all species are threatened consistently meet the EDGE criteria (see Supplementary Materials for these results).

## Reporting summary

Further information on research design is available in the Nature Portfolio Reporting Summary linked to this article.

## Data availability

The data underlying all results, figures and tables are available on Figshare (DOI: 10.6084/m9.figshare.22689793). Summary statistics of species, family and clade-level evolutionary history and EDGE scores

are provided as part of Supplementary Data 1. Species taxonomies were taken from Frost's Amphibian Species of the World (https://amphibiansoftheworld.amnh.org/), The Reptile Database (http://www.reptile-database.org/data/), BirdLife International (https://datazone.birdlife.org/species/taxonomy), the ASM Mammal Diversity Database (https://www.mammaldiversity.org/), and Fishbase (https://www.fishbase.se/search.php). Phylogenies were taken from VertLife (https://vertlife.org/data/) and the Fish Tree of Life (https://fishtreeoflife.org/). Extinction risk data were taken from the IUCN Red List version 2021.1 (https://www.iucnredlist.org/). Source data are provided with this paper.

## Code availability

Species-specific (terminal branch length, ED2, EDGE2) and clade-level (total evolutionary history and expected loss) scores were generated using the EDGE2 code available here: https://github.com/rgumbs/EDGE2.

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

## Acknowledgements

Through J.R. and W.D.P., this study is an output of the Georgina Mace Centre for the Living Planet at Imperial College London. R.G. was funded by the NERC Science and Solutions for a Changing Planet Doctoral Training Programme (grant number NE/L002515/1), the CASE component of which was funded by the Zoological Society of London.

## Author contributions

RG, JR, NRO, WJ, WDP, MH, CL, MB, CLG conceived the study. RG, OS, JR, MH, NRO, CL designed the analyses. RG, RB, OS, DK, CL collected and curated data. RG and OS conducted the analyses. JR, MB, MH, NRO, WJ, FF, SP, STT, BT, CLG provided technical support and conceptual advice. JR, NRO, MH supervised the study. RG wrote the paper, with contributions from all authors.

## Competing interests

The authors declare no competing interests.
