## [Peer Review File · Nature Communications]

Reviewers' Comments:

Reviewer #1:

Remarks to the Author:

This is a nice study of the impact of human-driven extinctions on jawed vertebrate evolutionary history. The study showed an updated EDGE2 values for all jawed vertebrates and provide a detail analyses of the evolutionarily distinct lineages. This study provides important insights for species conservation. There is a high importance to take into account PD in international conservation assessments (IPBES and CBD). This study is in line with this needs.

Overall, I think the study is very well-conducted and the methodology is strong. However, the study is quite long with a lot of results (maybe some are not necessary), and the authors could focus a little bit more their writing and results. I also believe the methodology could be better explained in some sections (see my other comments).

1) In a conservation context, I am wondering whether the IUCN500 scenario is really useful here. Species' protection is very difficult to conduct and I am not sure that outcome over 500 years (given the increasing pressure on species) are relevant and not associated with too much uncertainty. I would argue that only 50 years scenarios is sufficient here. Instead of the IUCN500, the absolute value of evolutionary history of each group could be represented in Fig 1A. Maybe I misunderstand but the EDGE is calculated by weighting extinction risk also over a 50years extinction scenarios. It seems from the legend of the figure and reading the ms that EDGE is not a 50 yrs extinction scenario, could the authors please clarify?

2) L116 The evolutionary history represented by each species within group is also a very important information, this is not sufficiently developed within the ms. Figure S2 could be included in the main ms. Given the role of species richness in PD, I would also expect a sensitivity analysis of the evolutionary history loss controlled by species richness. Those results could be highly relevant to emphasize some priority groups (hosted a higher PD than expected given their respective SR).

3) I am wondering if the evolutionarily distinct lineages calculation is not redundant with the EDGE2 calculation, which also takes into account the risk of extinction of all descendants' species of that branch. Theoretically, EDGE score will be higher for a species that is threatened and monotypic compared with species that is not distinct and with non-threatened closed species. I am not sure that the advantages of the two calculations are clearly expressed in the ms, and why both are necessary? If both results bring complementary information, I would suggest to the authors to compare both lists to provide conservation priorities

Minor comments:

Within the ms, the authors used EDGE2 using the new method, but sometimes they used EDGE, so this could be a little bit confusing, please use one or the others or define from the very beginning that EDGE is used for simplicity. Moreover, L112 the authors "calculated EDGE scores" and L133 they explained how they calculated EDGE scores using EDGE2 protocol, it gives the impression that they did not apply the same methodology for Figure 1 (L112). Please clarify. Also in the method both EDGE2 (L551) and EDGE (L554, L575). Both terms are used and this is very confusing.

L65 I appreciate the author acknowledge the policy context of the work L56-64 but, they did not emphasize the scientific literature background (e.g., <https://www.nature.com/articles/s41467-021-23861-y>)

L89- I am not surprised that the ray finned fish contribute the largest amount give the total number of species of this group. Did the authors considered to control by species richness?

Maybe one of the conservation issue would be to maximize the evolutionary history conserve for a "minimal" number of species. In this context, it would be important to explore the role of SR on those

values.

L84-85 "for three quantifications of extinction risk derivate", this sentence alone without the context cannot be understand by the readers. Please give more details.

L99: I don't understand how birds are at risk of losing 7% when the range is from 3% up to 6%.

L137-138 seems a bit repetitive with L116-120. I am not sure which new information is provided here.

L 488 The EDGE2 formula should be presented in the ms, given the central role of this formula, the readers should be able to understand how edge values were calculated directly from the ms.

Reviewer #2:

Remarks to the Author:

Dear Editor and authors,

I reviewed the manuscript "Global conservation status of the jawed vertebrate Tree of Life", which presents the most up-to-date and comprehensive prioritization of species and lineages to protect the vertebrate tree of life. This manuscript builds on past efforts to prioritize species that represent unique evolutionary branches of the tree of life, namely the EDGE of Existence, but advances some steps forward by integrating an updated algorithm (EDGE2), including for the first-time bony fishes in the prioritization, and integrating almost all the vertebrate tree of life in a single approach. In this sense, the study provides an important tool for future conservation efforts trying to reduce the loss of entire branches of the evolutionary history on Earth. The data used for generating the indices are the most up-to-date and comprehensive possible, and the authors use appropriate methods to secure the robustness of the analysis whenever necessary. The writing is good and tackles the necessary topics to understand the background and results of the work. In summary, it seems this manuscript was already revised before and presents all the necessary components to be a great contribution to conservation biology. I am available for any further clarifications.

Best regards,

Bruno Eleres Soares, Ph.D.

University of Toronto-Scarborough

Reviewer #3:

Remarks to the Author:

Dear authors,

I was pleased to review the manuscript entitled 'Global conservation status of the jawed vertebrate Tree of Life' by Gumbs and collaborators submitted for consideration at Nature Communications. This manuscript provides a significant contribution to the understanding of human-driven extinction risks faced by jawed vertebrates. The use of multiple analytical approaches allows for a comprehensive assessment of the threats to evolutionary history and identifies potential strategies to conserve these vital components of the Tree of Life. The findings underscore the need for immediate conservation actions to prevent the irreversible loss of jawed vertebrate diversity and the evolutionary heritage they represent. Congratulations on the study.

We thank the editor and reviewers for their comments. We have addressed the editorial requests, updating Figure 3 to a boxplot and submitting up to date reporting forms. We have addressed all of the reviewer points below. Most notably, we have: i) moved the analyses on identifying candidate EDGE species through taxonomy alone, and the associated methods and discussion text, to the supplementary materials; ii) added a new figure at the request of reviewer 1 to show the species-specific contributions to PD and threatened PD (Figure 2); iii) clarified vague or confusing points throughout; and iv) removed redundant or unnecessary text throughout. Reviewer comments are in black and our responses are italicised and in blue.

REVIEWER COMMENTS

Reviewer #1 (Remarks to the Author):

This is a nice study of the impact of human-driven extinctions on jawed vertebrate evolutionary history. The study showed an updated EDGE2 values for all jawed vertebrates and provide a detail analyses of the evolutionarily distinct lineages. This study provides important insights for species conservation. There is a high importance to take into account PD in international conservation assessments (IPBES and CBD). This study is in line with this needs.

We thank the reviewer for their feedback.

Overall, I think the study is very well-conducted and the methodology is strong. However, the study is quite long with a lot of results (maybe some are not necessary), and the authors could focus a little bit more their writing and results. I also believe the methodology could be better explained in some sections (see my other comments).

We have moved the discussion on EDGE families and candidate EDGE species to the supplementary and revised the discussion to reduce the word count, and provided better explanation of the methods, including incorporating formulae and a table outlining specific extinction risk quantifications.

1) In a conservation context, I am wondering whether the IUCN500 scenario is really useful here. Species' protection is very difficult to conduct and I am not sure that outcome over 500 years (given the increasing pressure on species) are relevant and not associated with too much uncertainty. I would argue that only 50 years scenarios is sufficient here. Instead of the IUCN500, the absolute value of evolutionary history of each group could be represented in Fig 1A. Maybe I misunderstand but the EDGE is calculated by weighting extinction risk also over a 50years extinction scenarios. It seems from the legend of the figure and reading the ms that EDGE is not a 50 yrs extinction scenario, could the authors please clarify?

The EDGE2 quantification is a weighting that is set at the same CR value as the 50-year conversion of Red List categories (0.97) but then uses an approach where extinction risk

halves with every decrease in Red List category, following the EDGE2 calculation and the PD indicator. We have added a table outlining the values to the methods (Table 1).

Plotting the total PD on Fig. 1A would cause the y-axis to jump from a maximum of 60 billion years (GY) to 340 GY, due to the large PD of fish, rendering several groups as lines at the bottom without logging the data (much like crocodylians already are on the current plot). We did not want to log the plot as it provides a sense of the varying magnitudes at risk of being lost to contrast against the proportional loss in panel B. However, we agree that showing the total PD is useful and so have added text to the results to highlight this and a supplementary figure that shows the expected loss and total PD on a single boxplot as suggested (Supplementary Fig. 1).

While we agree that the 500-year time horizon for quantifying extinction risk has the least utility for conservation of the three, we have included it as it provides an additional way of incorporating the inherent uncertainty we face in measuring expected loss across large phylogenies with an uncertain future (“if it’s even worse than we imagined, what could it look like?”). We have highlighted that we used three quantifications of extinction risk to explore the outcomes under differing extinction risk severities (lines 84-90 in the tracked change manuscript), and now open the results saying:

“Conversions of IUCN Red List categories to quantitative extinction risks are available for various time horizons from 50 to 500 years into the future^{3,13,37,38}. We used three quantifications of extinction risk—the 50-year and 500-year time horizons from Mooers et al.³⁷ and the ‘EDGE2’ quantification from Gumbs et al.¹³ (see methods)—to calculate the total and the proportion of jawed vertebrate evolutionary history (i.e., PD) expected to be lost under different severities of extinction risk.”

and added a table in the methods with the values for these (Table 1).

2) L116 The evolutionary history represented by each species within group is also a very important information, this is not sufficiently developed within the ms. Figure S2 could be included in the main ms. Given the role of species richness in PD, I would also expect a sensitivity analysis of the evolutionary history loss controlled by species richness. Those results could be highly relevant to emphasize some priority groups (hosted a higher PD than expected given their respective SR).

We have increased the attention on the evolutionary history represented per species in a clade by moving supplementary figure 2 to the main text (now Fig. 2) now with two panels to show what the reviewer suggested: expected PD loss of the clade controlled by species richness and total PD of the clade controlled by species richness, and added text on this to the results (lines 131-157). Where we now say:

“The amount of evolutionary history embodied by each species in a clade (measured as the total evolutionary history of the clade divided by the number of species) generally decreases as the species richness of the clade increases (Spearman’s rank correlation: = -0.67, $p = 0.08$; Fig. 2a). Sharks and rays embody the most evolutionary history per species across all jawed vertebrate clades (32 MY per species; ANOVA with Tukey HSD: $p < 0.001$ vs. all other clades; Fig. 2a). They are followed by crocodylians (24.5 MY per species) and testudines (18.6 MY per species). Amphibians embody 17.7 MY of evolutionary history per species despite comprising an order of magnitude more species than testudines, and amphibian evolutionary history per species is more than double that of the similarly speciose

mammals (6 MY per species) and more speciose birds (8.2 MY per species; Fig. 2a). Similarly, ray-finned fish embody significantly more evolutionary history per species (10.1 MY per species) than the much smaller mammal and bird clades ($p < 0.0001$ for all comparisons).

The crocodylians, the least speciose clade in the study, also embody more threatened evolutionary history per species (5 MY; measured as the total threatened evolutionary history of the clade divided by the number of species; Fig. 2b) than any other clade ($p < 0.0001$), followed by the second and third least speciose clades: testudines (4.8 MY per species) and chondrichthyans (4.7 MY per species). Amphibians embody significantly more threatened evolutionary history per species than mammals (3.1 MY vs. 0.6 MY; $p < 0.0001$). Lepidosaurs embody significantly more threatened evolutionary history per species than both mammals and the similarly speciose birds (1.9 MY vs 0.6 MY; $p < 0.0001$) and the more species-rich ray-finned fish (1.9 MY vs 1.2 MY; $p < 0.0001$; Fig. 2b)."

We have reinforced our existing emphasis on these groups where there is a high species-specific responsibility for the conservation of overall PD in the discussion (lines 408-419), where we now say:

"Despite having marginally lower levels of overall threatened evolutionary history than crocodylians or testudines, amphibians and sharks and rays embody particularly large amounts of threatened evolutionary history relative to their species richness (Fig. 2b), indicating the loss of species from these clades will likely result in larger losses of evolutionary history on average. Lepidosaurs have a proportion of threatened evolutionary history comparable to chondrichthyans, despite having a smaller proportion of threatened species, due to the large amount of total and threatened evolutionary history embodied by each lepidosaur species on average, which is significantly greater than that of the comparably speciose birds (Fig. 2). This large amount of per-species evolutionary history embodied by lepidosaurs, relative to similarly speciose groups, is driven by their ancient crown age (~240 MYA, compared with ~150 MYA for birds), relatively old family-level phylogenetic divergences (Supplementary Fig. 5), and relatively long terminal branches (Supplementary Fig. 2)."

We also noticed that our initial calculations of PD per species were actually those of threatened PD per species and have updated the text and figures accordingly.

3) I am wondering if the evolutionarily distinct lineages calculation is not redundant with the EDGE2 calculation, which also takes into account the risk of extinction of all descendants' species of that branch. Theoretically, EDGE score will be higher for a species that is threatened and monotypic compared with species that is not distinct and with non-threatened closed species. I am not sure that the advantages of the two calculations are clearly expressed in the ms, and why both are necessary? If both results bring complementary information, I would suggest to the authors to compare both lists to provide conservation priorities

The evolutionarily distinct lineages calculation follows the original IUCN formulation in 2012 and is important to report upon to track our progress towards IUCN resolutions. We now see that this section was confusing and have added clarity around the family-level EDGE calculations and also added the comparison suggested by the reviewer.

The family-level EDGE lineages analyses are designed to complement the species-level EDGE approach in two ways:

1) by highlighting regions of the tree (in this case families) that are at risk despite their constituent species not ranking at the top of the EDGE species lists - for example, a family of 5 Critically Endangered species on relatively short terminal branches will have lower EDGE scores than 5 Critically Endangered species from monotypic families with long terminal branches, but identifying these families where there are 5 threatened species that are not necessarily all high EDGE can help highlight regions of the tree where there is a danger of cascading effects of phylogenetic clumping of extinction risk. This is because EDGE2 assumes independent extinction across all species but in practice extinction may not be independent, especially among closely related species, the risk of losing a clade of 5 closely related and threatened species might be a lot larger than implied by EDGE2.

2) to enable the meaningful EDGE prioritisation of groups where species-level data is poor or research is often focused on higher taxonomic levels (e.g. for flowering plants / invertebrates). We believe it is important to show that EDGE can be applied at these taxonomic levels to enable its wider application across the tree of life.

We have added a comparison of the species vs family lists where we correlate: 1) the EDGE Lineage rank of a family with its average species EDGE rank, which shows a strong positive correlation ($\rho = 0.98$); and 2) the average species rank against the species richness of the family ($\rho = 0.46$) which shows that larger families contain lower ranking EDGE species on average. We have added a supplementary figure which compares average species EDGE rank and family EDGE rank, and also compares lowest EDGE rank of the species in the family against the family rank to highlight how conserving even lower ranked EDGE species can safeguard deeper branches (Supplementary Fig. 4). We now say (lines 281-290):

“To complement the species-level EDGE prioritisation, we calculated the mean EDGE score for each vertebrate family to delineate EDGE Lineages, i.e., the most evolutionarily distinct families where all assessed species are threatened with extinction. These families represent deep branches of the Tree of Life that are at the most acute risk and may be overlooked by the species-level approach, as they are descended by multiple species, and this approach can be applied to taxonomic groups where conservation research is often targeted at higher taxonomic ranks (e.g., plants⁴). Overall, the rank of EDGE Lineages is strongly positively correlated with the mean EDGE rank of their constituent species ($p = 0.98$, d.f. = 1,125, $p < 0.0001$), and the mean species EDGE rank of a family increases (i.e., lower priority) as the species richness of the family increases ($\rho = 0.43$, d.f. = 1,125, $p < 0.0001$; Supplementary Fig. 4).”

We have also added extra information on an example where the families approach adds value: for large families of threatened primates (lines 306-317), where we now say:

“Despite mammals being absent from the top 25 EDGE jawed vertebrate species and being underrepresented in the top 1% of EDGE scores (Fig. 4), there is a mammal family in the top 25 EDGE Lineages: Daubentoniidae (the Aye-aye, *Daubentonia madagascariensis*). Mammals have the highest number of fully-threatened families of all groups (Supplementary Table 1) and consequently comprise one-third of EDGE Lineages (21 of 63), more than any other clade. The four most speciose EDGE Lineages are primate families: sportive lemurs (*Lepilemuridae*; 26 spp.); true lemurs, ruffed lemurs and bamboo lemurs (*Lemuridae*; 21 spp.); gibbons (*Hylobatidae*; 20 spp.); and sifakas, woolly lemurs and the Indri (*Indriidae*; 19 spp.). However, mammalian families are significantly younger in age, compared with amphibians, ray-finned fish, chondrichthyans, lepidosaurs and testudines ($p < 0.0001$) and younger, as a proportion of clade age, than families in all other clades excluding ray-finned fish ($p < 0.0001$ vs; Supplementary Fig. 5).”

We have moved the candidate EDGE analysis (“Using taxonomy to identify candidate EDGE species” section) to the supplementary material to reduce word count and improve clarity.

Minor comments:

Within the ms, the authors used EDGE2 using the new method, but sometimes they used EDGE, so this could be a little bit confusing, please use one or the others or define from the very beginning that EDGE is used for simplicity. Moreover, L112 the authors “calculated EDGE scores” and L133 they explained how they calculated EDGE scores using EDGE2 protocol, it gives the impression that they did not apply the same methodology for Figure 1 (L112). Please clarify. Also in the method both EDGE2 (L551) and EDGE (L554, L575). Both terms are used and this is very confusing.

Agreed that this is confusing. We have clarified throughout that we are using only EDGE2 methodology, which we refer to as EDGE scores for simplicity throughout.

L65 I appreciate the author acknowledge the policy context of the work L56-64 but, they did not emphasize the scientific literature background (e.g., <https://www.nature.com/articles/s41467-021-23861-y>)

We have added the suggested citation where relevant (relating to the assessment of terrestrial vertebrates, line 61) along with two others for invertebrates. We feel that this, along with the other 12 citations provided from lines 61-64, adequately captures the scientific literature underpinning this research.

L89- I am not surprised that the ray finned fish contribute the largest amount give the total number of species of this group. Did the authors considered to control by species richness? Maybe one of the conservation issue would be to maximize the evolutionary history conserve for a “minimal” number of species. In this context, it would be important to explore the role of SR on those values.

This is addressed by Panel B in figure 1 (% evolutionary history loss, which puts all loss on a comparable scale), and the new figure 2 and associated text (see earlier response).

L84-85 “for three quantifications of extinction risk derivate”, this sentence alone without the context cannot be understand by the readers. Please give more details.

We have clarified this in the text, where we now say:

“Conversions of IUCN Red List categories to quantitative extinction risks are available for various time horizons from 50 to 500 years into the future^{3,13,37,38}. We used three quantifications of extinction risk—the 50-year and 500-year time horizons from Mooers et al.³⁷ and the ‘EDGE2’ quantification from Gumbs et al.¹³ (see methods)—to calculate the total and the proportion of jawed vertebrate evolutionary history (i.e., PD) expected to be lost under different severities of extinction risk.”

(line 85-91).

L99: I don't understand how birds are at risk of losing 7% when the range is from 3% up to 6%.

Sorry for the confusion – that is not the range, it is the 50 and 500 timeframe values in the parentheses We have clarified this in the text (line 105-112):

“In terms of the proportion of evolutionary history at risk across clades, testudines (turtles and tortoises) are at risk of losing 26% of their total evolutionary history under the EDGE2 extinction risk quantification (20%-34% for 50-year and 500-year time horizons), crocodylians 21% (50-year: 16%; 500-year: 22%), and amphibians 17% (50-year: 15%; 500-year: 26%; Fig. 1b). Despite having the greatest total threatened evolutionary history, ray-finned fish are at risk of losing only 12% (50-year: 11%; 500-year: 18%), lower than lepidosaurs (EDGE2: 15%; 50-year: 13%; 500-year: 22%) and chondrichthyans (EDGE2: 15%; 50-year: 12%; 500-year: 22%). Mammals are at risk of losing 10% (50-year: 7%; 500-year: 14%) and birds 7% (50-year: 3%; 500-year: 6%).”

We also already note this specific case in the discussion (line 384-392):

“For birds, the proportion of threatened evolutionary history under the EDGE quantification of extinction risk is comparable to that under the pessimistic 500-year scenario. This pattern is driven by the large proportion of Least Concern bird species (77%, the highest of all clades by 25%), which have a higher extinction risk value under the ‘EDGE2’ quantification¹³ (0.06) than the 500-year quantification (0.0005), and the relatively small proportion of EN (4%) and CR (2%) species, which have lower extinction risk values under the EDGE2 quantification (0.485 and 0.97 versus 0.996 and 1 under the 50-year quantification³⁷, respectively). Estimations of threatened evolutionary history using the EDGE2 quantification are therefore more pessimistic for LC species, relative to the 500-year quantification, but more optimistic for EN species.”

L137-138 seems a bit repetitive with L116-120. I am not sure which new information is provided here.

Lines 116-120 is focused on PD per species (no extinction risk included) for each clade, whereas 137-138 is focused on EDGE scores per species (mean EDGE), which is PD weighted by extinction risk. This section has now been altered as part of an earlier suggestion by the reviewer to emphasise both of these aspects (now lines 131-157 and copied above for the earlier response).

L 488 The EDGE2 formula should be presented in the ms, given the central role of this formula, the readers should be able to understand how edge values were calculated directly from the ms.

You're right! We have added formulae for PD, expected PD loss and the PD indicator to the methods (lines 574-595):

“To produce the total evolutionary history of a clade, we calculated the total phylogenetic diversity (PD) by summing all branch lengths of the phylogenetic tree:

$$PD(k) = \sum_{i=1}^b L_i$$

Where k represents the phylogenetic tree, b gives the total number of phylogenetic branches in k , i is an index identifying individual branches from k , and L_i is the length of branch i where $1 \leq i \leq b$. To estimate the amount of evolutionary history at risk, we follow Faith⁸⁸ by calculating the expected PD loss:

$$\text{expected PD loss}(k) = \sum_{i=1}^b L_i \left(\prod_{j=1}^{n_i} q_{ij} \right)$$

Where n_i gives the number of descendant species from branch i , and q_{ij} gives the probability of extinction for descendant species j from branch i where $1 \leq i \leq b$ and $1 \leq j \leq n_i$. This approach calculates the amount of evolutionary history from each phylogenetic branch we expect to lose, given the extinction probability of all descendant species of that branch. For example, if we consider a branch descended by two species, both of which have a probability of extinction of 0.5, there is a 0.5×0.5 (0.25) probability of losing the branch's evolutionary history, and thus the expected loss is 25% of the branch's evolutionary history. When this is calculated and summed for all branches in the phylogenetic tree, we have the total amount of evolutionary history that we stand to lose for the entire clade, and from this we can derive the proportion of total evolutionary history expected to be lost. We calculated the percent of evolutionary history at risk as:

$$\text{Prop. thr. evo. hist.}(k) = \frac{\text{Expected PD loss}(k)}{PD(k)} \times 100$$

This proportion of the total evolutionary history of a clade we expect to lose underpins the Phylogenetic Diversity indicator adopted by the CBD's GBF^{17,36}.

And added the formula for EDGE2 (lines):

“We followed Gumbs et al.¹³ by calculating EDGE2 scores for species i as:

$$EDGE2_i = ED2_i \times GE2_i$$

Where the EDGE2 of species i is a product of its ED2 and GE2 scores. GE2 scores are the extinction risk weightings outlined in Table 1, and ED2 is calculated for species i as:

$$ED2_i = TBL_i + \sum_{j \in A(i)} \left(L_j \times \prod_{h \in C_j \setminus \{i\}} GE2_h \right)$$

where $ED2_i$ consists of the TBL_i (terminal branch length of species i) plus another component corresponding to the expected contribution of shared internal branches to future terminal branch length i . The set $A(i)$ contains all branches ancestral to species i but excluding the terminal branch. As before, L_j gives the length of branch j . The set C_j contains all descendant species of internal branch j . $C_j \setminus \{i\}$ uses mathematical set minus notation to indicate contains all descendant species of internal branch j excluding species i . $GE2_h$ is the extinction risk weighting of species h . $ED2_i$ is therefore species i 's expected terminal branch length at some future time after all other species in the tree survive or become extinct based on their associated $GE2_h$ values (i.e., product of all $GE2_h$)¹³."

Reviewer #2 (Remarks to the Author):

Dear Editor and authors,

I reviewed the manuscript "Global conservation status of the jawed vertebrate Tree of Life", which presents the most up-to-date and comprehensive prioritization of species and lineages to protect the vertebrate tree of life. This manuscript builds on past efforts to prioritize species that represent unique evolutionary branches of the tree of life, namely the EDGE of Existence, but advances some steps forward by integrating an updated algorithm (EDGE2), including for the first-time bony fishes in the prioritization, and integrating almost all the vertebrate tree of life in a single approach. In this sense, the study provides an important tool for future conservation efforts trying to reduce the loss of entire branches of the evolutionary history on Earth. The data used for generating the indices are the most up-to-date and comprehensive possible, and the authors use appropriate methods to secure the robustness of the analysis whenever necessary. The writing is good and tackles the necessary topics to understand the background and results of the work. In summary, it seems this manuscript was already revised before and presents all the necessary components to be a great contribution to conservation biology. I am available for any further clarifications.

Best regards,
Bruno Eleres Soares, Ph.D.
University of Toronto-Scarborough

We thank the reviewer for their positive appraisal.

Reviewer #3 (Remarks to the Author):

Dear authors,

I was pleased to review the manuscript entitled 'Global conservation status of the jawed vertebrate Tree of Life' by Gumbs and collaborators submitted for consideration at Nature Communications. This manuscript provides a significant contribution to the understanding of human-driven extinction risks faced by jawed vertebrates. The use of multiple analytical approaches allows for a comprehensive assessment of the threats to evolutionary history and identifies potential strategies to conserve these vital components of the Tree of Life. The findings underscore the need for immediate conservation actions to prevent the irreversible loss of jawed vertebrate diversity and the evolutionary heritage they represent. Congratulations on the study.

We thank the reviewer for their positive appraisal of the manuscript.

Reviewers' Comments:

Reviewer #1:

Remarks to the Author:

I thank the authors for their revised version and I am mostly satisfied on how the authors addressed my previous comments. I still have two minor comments related to the justification of the study :

- The authors briefly justified how PD could be useful as an indicator of biodiversity, which I agree but if the readers is not familiar with PD measure, I think some concrete examples on how PD contributes to the ecosystem productivity, or human well being could be useful here.

-In the discussion, I believe one of the important limitation/issue that is not mentioned is the rationale to rank species among jawed vertebrates clades. The conservation measures between terrestrial or marine species or mammals or turtles or birds, could be very different and the results presented here are highly dependant on which clades are considered. Why grouping all those clades together and not mammals only, vs birds and ranking those species among the same taxa, I think this should be justified and at least discussed how this could affect the ranking. For instance, the ranking could be used to justify that mammals are finally not very important to preserve PD, so some limitations could be useful (e.g, l 429 could be more nuanced for instance)

We thank the editor and reviewer for their comments. We have made the required changes and provide our point-by-point response below in italics.

Reviewer #1 (Remarks to the Author):

I thank the authors for their revised version and I am mostly satisfied on how the authors addressed my previous comments. I still have two minor comments related to the justification of the study :

- The authors briefly justified how PD could be useful as an indicator of biodiversity, which I agree but if the readers is not familiar with PD measure, I think some concrete examples on how PD contributes to the ecosystem productivity, or human well being could be useful here.

Response: We have added more detail to the section, where we now say:

“Evidence suggests that Amazonian forests that contain greater evolutionary history have higher wood productivity¹¹, and selecting sets of species to maximise evolutionary history can effectively capture species with known uses by people across the world’s plants¹² and birds¹³.” (lines 49-52)

-In the discussion, I believe one of the important limitation/issue that is not mentioned is the rationale to rank species among jawed vertebrates clades. The conservation measures between terrestrial or marine species or mammals or turtoises or birds, could be very different and the results presented here are highly dependant on which clades are considered. Why grouping all those clades together and not mammals only, vs birds and ranking those species among the same taxa, I think this should be justified and at least discussed how this could affect the ranking. For instance, the ranking could be used to justify that mammals are finally not very important to preserve PD, so some limitations could be useful (e.g, l 429 could be more nuanced for instance)

Response: We agree that this is an important point to make and, whilst we state in the methods (line 546-549) that we identify EDGE species for each clade, this was not made clearer in the discussion. We have updated this to now say:

“The high proportion of mammalian families comprising jawed vertebrate EDGE lineages—despite their relative absence from the top 1% of species EDGE scores (Fig. 3)—suggests that while individual mammal species may not represent incredibly large amounts of threatened evolutionary history, the phylogenetic clumping of extinction risk across entire families is nevertheless threatening deep branches of the mammal phylogenetic tree. It is for this reason that EDGE species—the most evolutionarily distinct and threatened species from a given clade identified to guide priority setting—are identified independently for each vertebrate class^{62,63}; to ensure clade-specific priorities for conserving evolutionary history are not overlooked when focusing on the wider Tree of Life. This becomes ever more important as EDGE scores are calculated for across an increasing number of clades across the entire Tree of Life. Whilst we present all jawed vertebrate species combined here to highlight priorities across the

entire jawed vertebrate Tree of Life, we also determined whether each species is an EDGE species within its respective clade to guide priority setting (Supplementary Data 1). EDGE Lists for all clades are maintained and updated by the Zoological Society of London's EDGE of Existence (www.edgeofexistence.org) to support conservation decision making.” (lines 359-372)